# Experimental Procedure for the Metrological Characterization of Time-of-Flight Cameras for Human Body 3D Measurements

**DOI:** 10.3390/s23010538

**Published:** 2023-01-03

**Authors:** Simone Pasinetti, Cristina Nuzzi, Alessandro Luchetti, Matteo Zanetti, Matteo Lancini, Mariolino De Cecco

**Affiliations:** 1Department of Industrial and Mechanical Engineering, University of Brescia, Via Branze 38, 25125 Brescia, Italy; 2Department of Industrial Engineering, University of Trento, Via Sommarive, 9, 38123 Trento, Italy; 3Department of Medicine and Surgery, Radiology and Public Health, University of Brescia, Viale Europa 11, 25123 Brescia, Italy

**Keywords:** machine vision, human-body reconstruction, time-of-flight, metrological characterization, Azure Kinect, Basler Blaze 101

## Abstract

Time-of-flight cameras are widely adopted in a variety of indoor applications ranging from industrial object measurement to human activity recognition. However, the available products may differ in terms of the quality of the acquired point cloud, and the datasheet provided by the constructors may not be enough to guide researchers in the choice of the perfect device for their application. Hence, this work details the experimental procedure to assess time-of-flight cameras’ error sources that should be considered when designing an application involving time-of-flight technology, such as the bias correction and the temperature influence on the point cloud stability. This is the first step towards a standardization of the metrological characterization procedure that could ensure the robustness and comparability of the results among tests and different devices. The procedure was conducted on Kinect Azure, Basler Blaze 101, and Basler ToF 640 cameras. Moreover, we compared the devices in the task of 3D reconstruction following a procedure involving the measure of both an object and a human upper-body-shaped mannequin. The experiment highlighted that, despite the results of the previously conducted metrological characterization, some devices showed evident difficulties in reconstructing the target objects. Thus, we proved that performing a rigorous evaluation procedure similar to the one proposed in this paper is always necessary when choosing the right device.

## 1. Introduction

Since the fourth industrial revolution started, industrial manufacturing technologies have evolved rapidly. Robots and machines are now equipped with intelligence and sensing devices such as cameras to see their environment. This is especially important when the machine operates alongside human workers. Therefore, the human body reconstruction capability of the camera is necessary for safety reasons and to guarantee the correct execution of detection and monitoring software. Typically, RGB cameras are the common choice for most industrial applications due to their competitive prices and low computational complexity. However, they lack information about the actual dimension of the environment, the relative distance of objects and their volume. Therefore, for some applications, 3D cameras are best suited for the task.

Three-dimensional cameras have been intensively adopted for years as measurement systems in a wide variety of applications. The exploited technologies are structured light, stereoscopy, and time-of-flight [1,2]. Time-of-flight (ToF) cameras are optical devices that measure the distance of objects from the sensor based on the calculation of the elapsed time between the emission/reflection of a light source. Compared to stereoscopy-based devices, they are best suited for indoor environments due to the possible interference from direct natural light. However, the depth image obtained from these devices is affected by artifacts due to quantization in the range of low amplitudes. This issue affects ToF technology due to large operating distances and low reflectivity. Moreover, the presence of bias and other systematic errors is caused by anharmonic signals and overexposure [3,4]. Illumination conditions, such as the target’s color properties and material reflectivity, are external error sources that cause depth errors, which increase linearly with distance [5]. Other common artifacts are flying pixels (erroneous depth estimates that appear close to the edge discontinuities in depth data) and multipath errors [6,7].

Two types of ToF cameras are available on the market: consumer-end and industrial. The main difference lies in the communication protocol: a Gigabit Ethernet connection is adopted by industrial cameras in contrast to USB 3.0 available in consumer-end devices. Industrial cameras are specifically designed for harsh environments; hence, the sensor and its lenses are encapsulated in a high-protection cover adhering to the IP protection index standard. They are typically adopted to reconstruct objects with high accuracy in controlled illumination and temperature conditions to guarantee the best performance. As such, their performance is often unsatisfactory at high distances and for moving subjects. In contrast, consumer-end devices are best suited for human activities, also thanks to the gaming industry, which intensively adopted this technology to enhance their user experience. As a result, consumer-end devices are compact and easy to use. Their 3D reconstruction capabilities are worse in terms of the accuracy of small objects and unusual surface materials; however, they work well with moving subjects. The most famous consumer-end device is the one developed by Microsoft, which has been improved over the years. Since the release of the Microsoft Kinect v2 sensor [8] in 2013, this commercial device has been intensively used for research purposes due to its affordability and performance [9]. For example, it has been used in 3D reconstruction for object modeling [10,11,12,13] and indoor scenes [14,15] and mobile robots’ navigation and mapping [16,17,18,19]. The industrial applications include palletizing tasks [20,21], safety [22,23,24], teleoperation [25,26], human body detection and tracking [27,28,29], and gesture recognition tasks [30,31,32]. Healthcare applications involve gait analysis and elderly monitoring [33,34,35,36], the reconstruction of human body kinematics thanks to augmented and virtual reality software based on Kinect v2 [37,38]. However, Microsoft interrupted the production of Kinect v2 devices in favor of its new product Kinect Azure released in 2020. Currently, only a few works performed a metrological characterization of the new sensor. The work presented in [39] describes a set of experiments focused on gait analysis aimed at comparing the performance of the new Kinect Azure with the old Kinect v2 and a Vicon system. The authors of [40] performed a thorough characterization of the Kinect Azure sensor compared with its predecessors, Kinect v1 [41] and Kinect v2. They evaluated the three sensors testing their depth repeatability, noise-to-reflectivity, warm-up time, depth precision, reflectivity sensitivity, lens aberration, indoor versus outdoor performance, and flying pixel error. Finally, in [42], the authors explored the depth errors of Kinect Azure in comparison with Kinect v2.

To our knowledge, there is no standard experimental procedure to determine the typical error sources of this technology (namely the temperature influence and depth-related errors). As a result, researchers often come up with their own procedure resulting in non-comparable results and inconsistencies, spreading confusion among the scientific community. For example, without a careful characterization of ToF devices (or a document clearly stating their metrological properties), researchers may end up using the camera of choice without considering the intrinsic errors this technology implies, errors that can be corrected in post-processing if known and estimated beforehand. Working with depth devices without this knowledge may lead researchers to reach wrong conclusions in their own works, for example, when bias correction is not correctly estimated and corrected or when temperature influence is not considered when designing experiments. Another example is when ToF devices are adopted as monitoring systems in industrial workspaces, leading to incorrect depth estimation that may result in harmful behavior of robotic systems and machines or in healthcare applications when patients’ body volume is incorrectly estimated. The datasheet information is typically not enough in the case of particular tasks; thus, a metrological characterization is always needed to ensure the success of the application. Therefore, our first contribution is the proposed experimental set-up and characterization procedure described in Section 2, Section 3 and Section 4.

Second, there is little to nonscientific literature that compares consumer-end ToF cameras with industrial ones because the two have very specific application fields that usually do not overlap. However, because of the Industry 4.0 paradigm, there are a plethora of new applications and requirements that challenge both worlds, thus making this comparison meaningful for both researchers and practitioners helping them choose the suitable device for their task. Therefore, the investigation conducted compares the performance and metrological characteristics of two industrial Basler ToF cameras with the new Kinect Azure in a set of experiments based on our previous work conducted on different ToF cameras [43].

Finally, considering the challenging task of human recognition and tracking in industrial environments [44], the devices have also been tested in two 3D reconstruction set-ups to evaluate the quality of their point clouds when used to reconstruct both geometrical objects and human body segments. This is, in fact, one of the tricky yet more interesting applications in which ToF cameras are typically involved as the metrological device to measure the objects’ dimension and volume. To this aim, the experimental procedure conducted to evaluate the correctness of both objects and human shapes dimensions is detailed in Section 5. The results are compared with the reconstruction obtained with a high-performing digitizer (gold standard).

## 2. Materials and Methods

ToF measurement is performed using a continuous-wave modulation based on the phase-shifting principle [45]. A periodic wave is emitted from the device and, after hitting an object, is sent back to the system. The resulting distance is calculated by analyzing the time between the emission of the wave and the corresponding received signal.

### 2.1. Specifications of the Evaluated Sensors

The cameras analyzed in this work are (i) Microsoft Kinect Azure (consumer-end), (ii) Basler Blaze 101 (industrial), and (iii) Basler ToF 640 (industrial). Table 1 details the main technical characteristics of the three devices.

It is worth noting that, compared to Kinect v2, Kinect Azure’s depth camera may be used in two modalities, which may be binned or unbinned: NFOV (narrow field-of-view) and WFOV (wide field-of-view). However, in the context of this work, only the NFOV unbinned modality was used to perform the tests since the binning operation is applied by the SDK. The depth sensor adopted by the camera is based on the one detailed in [46].

The Basler Blaze 101 ToF camera mounts a Sony DepthSense IMX556 sensor, which makes the camera more robust to natural light. According to the datasheet, the optimal operating range is 0.5–5.5 m, where a depth precision of ±5 mm is guaranteed.

The Basler ToF 640 industrial camera is based on pulsed ToF technology adopting a Panasonic CCD depth sensor. It is optimized to work in indoor environments due to the technology’s sensitivity to natural light. The optimal operating range is 0.5–5.8 m, where a depth precision of ±10 mm is guaranteed.

### 2.2. Evaluation of Error Sources and Performance

The error sources are subdivided into (i) systematic and non-systematic errors according to their nature [47], (ii) camera-dependent errors, (iii) and scene-dependent errors [48]. However, in this study, depth errors related to fixed pattern noise and internal light scattering have not been evaluated.

The study detailed in [8] shows a metrological characterization based on the Guide to the expression of Uncertainty Measurements [49] for the uncertainty analysis of 3D scene reconstruction. Moreover, in [43], a metrological characterization comparing the performance of Kinect v2 and Picoflexx was performed. As in [43], the error sources considered are (i) temperature-related errors and (ii) depth-related errors.

Temperature-related errors are systematic and camera-related errors that are relevant for ToF cameras because their technology is strongly affected by heat [3,4,5,9]. The internal temperature of the camera is due to the heating of the illumination unit and image sensor, which produces drifts in the depth measure. After reaching a stable temperature, the components’ characteristics do not change anymore. Consequently, it is suggested by the literature [48,50,51,52] to use ToF cameras after a warm-up time to obtain stable depth readings. However, the warm-up time is different according to the device, and some may compensate internally for this effect while others may not.

Except for the temperature-related errors, the other three error sources analyzed in this study are all depth-related: (i) depth amplitude error, (ii) depth distortion, and (iii) temporal error.

Depth amplitude is a systematic and camera-related error because the precision of depth measurements depends on the amount of light observed on each pixel [47]. Both underexposed and overexposed amplitudes may result in depth discrepancies since the illumination intensity is the highest at the center of the image and grows weaker around the borders. This effect leads to the overestimation of the depth values around the edges. However, if the object is very close to the emitter, the observed intensity may be higher, leading to pixel saturation.

Depth distortion is a systematic and camera-related error that occurs when the emitted light (typically a sinusoidal signal) is not generated correctly due to irregularities in the modulation process. Therefore, an offset is produced that only depends on the measured depth observed at each pixel. This error is assessed by comparing the depth measurements from devices with a reference ground truth distance [47,48].

Temporal errors are non-systematic camera-related errors that represent the depth variation of a pixel over time caused by measurement noise, which is more evident when the scene illumination is non-uniform, or the observed surface has low reflectivity [48]. This error also depends on the depth uniformity of the scene and on the integration time.

The overall measurement uncertainty was evaluated following the methodology presented in [8]. In addition, two other tests were performed to evaluate the performance of the sensors when used for (i) 3D reconstruction and (ii) body kinematic measurements. In the first experiment, the capability of the devices to reliably reconstruct and measure body segments is determined, while the second is aimed at evaluating the performance in human body segmentation.

### 2.3. Measuring Set-Up

Figure 1 shows a scheme of the setup. The three cameras were tested indoors at an environment temperature of 24 °C. An opaque white sheet of paper was used as the target for the acquisitions, mounted on a planar panel with verified planarity of 0.1 mm around the center area of 400 × 400 mm. Each camera was mounted at a fixed height of 1.5 m from the ground, and they were oriented perpendicularly with respect to the target using a tripod with integrated bubble levels. A laser rangefinder EXTECH DT40M with an accuracy of ±2 mm was used to verify the nominal position $D_{n}$ of the camera with respect to the target according to the experiment. The overall illumination of the scene was kept constant without the influence of natural light. 

## 3. Evaluation of Temperature-Related Errors

Referring to the set-up in Figure 1, in this experiment, the cameras were positioned one at a time at a constant nominal distance of $D_{n}=2\;m$, corresponding to half of their operating range. They were kept in the test area while turned off for at least 4 h before starting the experiments.

The experiment lasted 2 h for each camera. The images were acquired every 10 s at 30 fps. Frames belonging to a time window of 5 min were grouped together, resulting in groups of 30 frames. For each, a region of interest of 15 × 15 px centered around the central pixel of the image was extracted. Finally, for each group the mean depth value $\mu_{t}$ and the standard deviation $\sigma_{t}$ was computed, resulting in 24 data points $d_{t}$.

Depth measurements performed by ToF cameras are typically affected by a systematic error (bias). Thus, the measured depth must be corrected by this quantity to obtain correct readings. Bias estimation, in this case, was conducted using the following formula:(1)$$\mu_{t}=\frac{1}{5}\;\overset{24}{\underset{i=19}{\sum }}d_{t}\;b=\mu_{t}-D_{n}\;d_{t}^{*}=d_{t}-b$$

We chose to calculate $\mu_{t}$ as the mean of the last five data points $d_{t}$ because they are the most stable. Otherwise, the bias correction would have taken in consideration the great difference between the actual distance $D_{n}$ and the measured distance $D_{m}$ that is observed in the first data points before the warm-up kicks in (see Figure 2). Hence, the last five $d_{t}$ correspond to the time intervals after the warm-up time of the cameras. The mean depth values after bias correction $d_{t}^{*}$ are represented in Figure 2.

Surprisingly, Kinect Azure does not need to warm up to obtain a stable output, meaning that the depth readings are not correlated with the device temperature. This result contrasts with the work presented in [40], where the authors stated that the device needed at least 60 min of warm-up. However, in our experiment, Kinect Azure achieves stable values from the beginning without the need for a warm-up time. In fact, the bias correction for Kinect Azure could have been performed considering all the 24 data points $d_{t}$ in Equation (1) instead of the last five only since they were not significantly different. The experiment was performed twice to confirm this finding, highlighting the robustness of our measuring method in contrast with reference [40]. A possible reason to explain why we obtained a significantly different result is that the bias was not corrected in [40]. Furthermore, in our experiment, the relative distance camera-target was $D_{n}=2\;m$, while in [40], they used $D_{n}=0.8\;m$. Environmental characteristics could have impacted the measurement, for example, if the ambient illuminance interfered with Azure’s IR rays. In our case, even if the camera is used continuously for almost 120 min, the output is stable in terms of deviation from the actual distance, revealing a small wiggling behavior that leads to outputs that differ from the nominal distance of about −15 mm (around 1 mm after bias correction). Moreover, according to the experimental comparison in [7], the same behavior could be observed for Kinect v1, while the Kinect v2 mean depth data show a strong correlation with the device temperature (stable readings after 25 min). On the other hand, both the Basler Blaze 101 and the Basler ToF 640 cameras need a warm-up time of at least 50 min. It is worth noting that the deviation (shown as error bars) decreases with time for both Basler Blaze 101 and Basler ToF 640, but this could not be said for Kinect Azure, which shows stable deviation values uncorrelated with the device temperature. The reason for this different behavior could be that industrial cameras’ design considers the hazardous environment in which they could be deployed; hence, manufacturers may not account for temperature-related errors internally due to lack of space or design limitations.

## 4. Evaluation of Depth-Related Errors

Referring to the setup in Figure 1, in this set of experiments, the three cameras were placed at 15 nominal positions $D_{n}$ in the range of 1.7–4.5 m from the planar target, with a step of 0.2 m. For each nominal position, a total of 30 frames were recorded at 30 fps.

### 4.1. Depth Amplitude Errors Evaluation

To evaluate the depth amplitude of each sensor, it is necessary to analyze both the quality of the captured IR image and the corresponding depth map. In fact, depth accuracy is related to the amount of light received by each pixel because both under and over-exposed pixels result in depth errors.

On the one hand, IR images contain the light intensity of the reflected ray emitted from the ToF camera. Observing the images on the left in Figure 3, it is evident that the intensity of the captured light decreases around the corners of the images while it is at its maximum in the center. In the case of Kinect Azure, the dark circles appearing in the image are due to the visualization of the image but are not an error source, while the black patches on the corners of the image are due to the reduced field of view that happens in the NFOW modality and are absent in WFOV. In the context of this work, only the NFOV was evaluated since its characteristics are similar to those of the other cameras. In contrast, for the Basler Blaze 101 and the Basler ToF 640 cameras, the resulting intensity image is lighter and more uniform. Considering that the images in Figure 3 refer to a $D_{n}=1.7$ m, the differences in the pictures are due to (i) the optics mounted on the cameras, (ii) the different field of view, and (iii) the positioning of the tripod along the *x*-axis with respect to the target; hence, the scene is captured differently. This is also the reason why only a portion of the frame around the central pixel was considered for computing our analysis of the depth frames.

On the other hand, depth images contain the measured depth value corresponding to each pixel. In this work, they were obtained by analyzing the point cloud because resolution and data accuracy are higher, especially for the two industrial cameras. The resulting depth amplitude error is estimated from these depth images. Considering the experimental set-up described in Section 4, to each nominal camera position $D_{n}$ corresponds to a total of m=30 measured depth maps $D_{m}$. Hence, to each $D_{n}$ corresponds an average depth map $\mu_{m}$ computed by:(2)$$\mu_{m}=\frac{1}{m}\overset{m}{\underset{i=1}{\sum }}D_{i}$$

The resulting data must be centered around zero, so for each $D_{m}$ we first compute:(3)$$\varepsilon_{m}=\mu_{m}-D_{n}$$

Then, we calculate the average $\varepsilon_{m}$ as:(4)$$\bar{\varepsilon_{m}}=\frac{1}{m}\overset{m}{\underset{i=1}{\sum }}\varepsilon_{m}$$

Finally, for each nominal distance $D_{n}$ we obtain the error map (centered around zero) by removing the mean:(5)$$\varepsilon_{n}=\varepsilon_{m}-\bar{\varepsilon_{m}}$$

The images on the right in Figure 3 represent the depth amplitude error calculated following the abovementioned procedure. In the case of Kinect Azure, the depth amplitude error at $D_{n}=1.7$ m is mostly concentrated around ±5 mm. On the upper corners, we can observe peaks of −17 to +19 mm due to reflections occurring on the surface. Basler Blaze depth is denser thanks to an increased point cloud resolution; however, more than half of the image shows an overestimation of +5 to +13 mm, probably due to the higher amount of reflected light in this area. It is also possible to observe concentric waves due to the non-ideal wave generated by the camera’s emitter, which may be another error source causing this overestimation. The peak of −31 mm refers to a non-planar edge of the target. Finally, the depth amplitude error map of Basler ToF 640 shows that most pixels are underestimated by −5 to −19 mm, especially in the center area, while the angles of the target are overestimated by +20 to +32 mm. It is worth noting that even if the camera-target alignment was ensured before the acquisition, displacements might have occurred without us noticing. This is the reason why for the analysis described in the following Sections, we only consider a sub-portion of the depth map. Furthermore, we checked the depth amplitude error in a region of interest of 40 × 40 mm centered around the central pixel of the depth map. In this area, Kinect Azure’s error ranges from 2 to 5 mm, Basler Blaze 101′s ranges from 3 to 7 mm, and Basler ToF 640′s ranges from −10 to −6 mm. This region has been chosen to obtain comparable results with respect to the cameras’ datasheets.

### 4.2. Depth Distortion Evaluation

Depth distortion errors typically increase with distance. Therefore, it is important to evaluate the relative trend of depth values according to the nominal positions $D_{n}$ instead of the absolute values. Considering the experimental set-up described in Section 4 and the mathematical procedure detailed in Section 4.1 to compute $\varepsilon_{n}$, in this case each nominal camera position $D_{n}$ corresponds to a total of m=30 measured depth values $D_{m}$ calculated considering only the central pixel of each frame [43].

Figure 4 shows the resulting depth distortion error $\varepsilon_{n}$ of the three cameras for each nominal position. For Kinect Azure and Basler Blaze 101, the error is very low, showing a wiggling trend that is more prominent in the case of Azure Kinect. The depth distortion of Kinect Azure spans from −18 mm to 10 mm, which corresponds to the nominal distances of 2.7 m and 3.3 m, respectively. In the case of Basler Blaze 101, it has an almost constant behavior and spans from −11 mm to 9 mm, corresponding to the nominal distances of 1.7 m and 4.5 m, respectively. However, for Basler ToF 640, the distortion error is evident since the trend increases with distance. It ranges from −48 mm to 76 mm, corresponding to nominal distances of 1.7 m and 4.5 m.

### 4.3. Temporal Errors Evaluation

Temporal errors refer to fluctuations of the depth values due to measurement noise. Hence, considering the experiment described in Section 4.1, the temporal error is the standard deviation $\sigma_{m}$ over 30 frames for each measured position $D_{m}$:(6)$$\sigma_{m}=\sqrt{\frac{1}{m}\overset{m}{\underset{i=1}{\sum }}{\left(D_{i}-\mu_{m}\right)}^{2}\;}$$

The mean depth value $\mu_{m}$ appearing in this formula is obtained from Equation (2).

The deviation values are shown in Figure 5 according to the $\mu_{m}$ values. For all the cameras, the standard deviation $\sigma_{m}$ has an increasing trend. The results for both the Kinect Azure and Basler Blaze 101 cameras are similar. On the other hand, the standard deviation of the Basler ToF 640 camera is sometimes very high ($\sigma_{m}=3.75$ mm for $\mu_{m}=3429$ mm) or very low ($\sigma_{m}=0.61$ mm for $\mu_{m}=1786$ mm). The values of $\sigma_{m}$ are slightly higher for the Kinect Azure, which has values in a range of 0.81 mm and 3.19 mm versus the $\sigma_{m}$ computed for the Basler Blaze 101, which is in a range of 0.52 mm and 2.99 mm. Moreover, the trend lines (represented by the black dashed lines) are obtained as a linear regression over values $\sigma_{m}$. The regression coefficients $R^{2}$ are equal to 90% (slope 2.67, intercept 1.01 mm), 91% (slope 2.59, intercept 1.62 mm), and 48% (slope 2.64, intercept 0.52 mm) for the Kinect Azure, Basler Blaze 101, and Basler ToF 640, respectively. This shows that the performance of the Kinect Azure and the Basler Blaze 101 are equivalent, while highlighting that the Basler ToF 640 has higher variability compared to the other two cameras.

It is worth noting that Kinect Azure’s results adhere to those obtained in [42], which show equivalent standard deviation values that increase with distance.

### 4.4. Overall Depth Measurement Uncertainty Evaluation

To obtain the overall measurement uncertainty, a neighborhood of 20 × 20 px centered around the central pixel of the frame was extracted, and each depth value belonging to this squared region was considered, resulting in 400 data points per frame. The neighborhood is shown in red in Figure 3.

For each $D_{n}$, 30 frames were taken in the range of 1.7–4.5 m with a step of 0.2 m corresponding to 15 positions. The collected measurements are shown in Figure 6 for the three cameras and correspond to a total of around 168,000 data points (except the outliers, which were removed from the calculations). The results show that the Kinect Azure has very little dispersion since the measured depth values corresponding to the squared blue markers (Figure 6a) have small variation. However, there are few occurrences of incorrect readings observed only for nominal distances greater than 4.1 m due to some reflections occurring in the scene. This resulted in a mean deviation $\sigma_{0}$ equal to 13 mm, which is slightly lower than the value obtained for the Kinect v2 sensor resulting from the same experiment carried out in [43], which reported a $\sigma_{0}$ value equal to 18 mm. The dispersion observed in the case of the Basler Blaze 101 camera is even better, as shown by the reduced variability of the green diamond markers (Figure 6b). The sensor seems less prone to dispersion effects, and its $\sigma_{0}$ is equal to 6 mm. Nonetheless, a high number of incorrect readings appear for nominal distances greater than 3.1 m. Compared to Kinect Azure, this effect seems more generally distributed since the incorrect measures span a wider range ($D_{n}$ in a range of 3.1–4.5 m corresponding to a$\;D_{m}$ range of 0–0.8 m) than in the case of Kinect Azure ($D_{n}$ in a range of 4.1–4.5 m corresponding to a$\;D_{m}$ range of 0.8–1.6 m). In contrast, the Basler ToF 640 camera has higher dispersion (bottom image in Figure 6c) but it does not achieve incorrect depth values, resulting in a $\sigma_{0}$ of 13 mm, which is the lowest of the three. Moreover, the linear regression computed for the three devices and represented by the black dashed line in each plot highlights their measurement linearity. The corresponding $R^{2}$ coefficients are 99.98%, 100%, and 99.98% for the Kinect Azure, Basler Blaze 101, and Basler ToF 640 cameras, respectively. It is worth noting that the outliers observed were removed before performing the analysis.

## 5. Application Example: 3D Reconstruction

ToF devices are typically used indoors to perform a plethora of measurement tasks, for example, the relative distance monitoring between human subjects and moving machines [23] and object volume and size estimation [53]. Depth cameras are also extremely important for the healthcare sector since human body reconstruction and volume estimation are crucial for the gait analysis of patients with reduced mobility [35] and for the evaluation of particular diseases involving malformations of the body [54].

A careful metrological characterization of the ToF camera of choice following the procedure described in Section 2, Section 3 and Section 4 is needed to determine the error sources and to design the application set-up in order to obtain correct data. However, in the case of 3D reconstruction, the assessment of error sources may not be enough to determine which device is best suited for the task. Therefore, in this Section, we propose two experimental set-ups in which the three cameras of our choice (Kinect Azure, Basler Blaze 101, and Basler ToF 640) are compared in terms of 3D reconstruction capabilities.

### 5.1. Object Reconstruction

This experiment aims to evaluate the sensors’ capabilities to accurately reconstruct objects from the acquired point cloud. The cylindrical object used in [55] with an external radius of 122 mm was used as the measurement target. The cylinder was industrially produced, and its radius was measured with a caliber with a 0.01 mm resolution. Considering the field-of-view (FoV) and range of the cameras, the cylinder was placed at 15 positions spanning the FoV symmetrically. However, since the reconstruction performance may vary according to the positioning of the target with respect to the camera in the vertical direction, two set-ups were considered where the bottom end of the cylinder is placed (i) at 0.7 m from the floor (odd stations numbers), and (ii) at 1.5 m (even stations numbers) with the aid of an adjustable carrier. This results in the set-up shown in Figure 7, where the red dots represent odd stations, and the green dots represent even stations. Stations 19, 23, 25, and 29 were moved to allow the cylinder to fit inside the camera FoV (black dashed lines). It is worth noting that in this experiment, the aim was not to evaluate the multipath effect; hence, the cylinder was not positioned at floor height.

A total of 30 frames at 30 fps for each station were acquired for each camera. By inspecting the data, it resulted that only the point clouds acquired by Kinect Azure were of enough quality for further analysis because the point clouds of the other two cameras were too noisy to allow a proper reconstruction of the object. For the Basler ToF 640 camera, the point cloud is affected by mixed pixel errors and multipath making it impossible to properly detect the target. In the case of the Basler Blaze 101 camera, the object is sometimes reconstructed at different depth values than the carrier, leading to incorrect readings. This is probably due to the target shape since industrial cameras are usually optimized to work with planar surfaces. Therefore, the following analysis was conducted only on Kinect Azure data:Each point cloud was manually inspected to remove the elements of the scene not belonging to the cylinder. This was performed by applying a depth filter to cut off data outside the area of interest, thus obtaining only the point cloud of the cylinder.The camera performance was evaluated by comparing the external radius of the measured cylinder with respect to the nominal one of 122 mm. The measured external radius was estimated individually for each acquisition by analyzing the point cloud with MATLAB using a cylindrical fit provided by the software.For each station, the mean value over 30 frames of the external radius and the corresponding standard deviation was computed.

Figure 8 shows the measured external radius for the odd and even positions, respectively. The diameter of the colored circle represents the mean value $\mu_{d}\;$of the measured diameter plus the standard deviation value $\sigma_{d}$ (upper bound, *UB*). The diameter of the white hole represents the mean value $\mu_{d}\;$minus the standard deviation value $\sigma_{d}$ (lower bound, *LB*). If this subtraction results in values less than zero, no hole is drawn.
(7)$$UB=\mu_{d}+\sigma_{d}\;LB=\mu_{d}-\sigma_{d}$$

The resulting mean values of the external radius span from 140 mm to 100 mm; however, the standard deviation is very high for stations 1, 2, 6, and 19, and it is above average for stations 7, 9, 11, 11, 20, and 30. Considering the camera operative range in NFOV unbinned mode, it is evident that the best performance is achieved when the cylinder is positioned at 1.5 m from the floor (even stations). Moreover, for stations positioned at depth values higher than 3.5 m, the standard deviation of the measure is higher, especially for stations closer to the edges of the FoV (1, 2, and 6, which have values of 432, 218, and 115 mm, respectively). It is unclear why, at positions 19, 20, and 30, the standard deviation is higher with respect to the other values achieved for stations inside the range of 1.5–3.0 m (values above 100 mm versus an average standard deviation of 50 mm). The reason may be that these are near the FoV edges. In conclusion, this experiment shows that Kinect Azure performance in 3D reconstruction is better at the center of its FoV, corresponding to the central area of the set-up.

### 5.2. Human Body Reconstruction

This experiment has been designed to evaluate the human body reconstruction capabilities according to the procedure described in [42]. A 70 × 30 cm mannequin representing the human upper body with movable arms was adopted for the test. Angles between its body segments were measured and compared with a reference measure taken by a commercial 3D digitizer Konica Minolta VIVID-920 with a resolution of 640 × 480 px and depth accuracy of ±0.40 mm. This device was chosen as the gold standard because its accuracy is one degree higher than the average measured depth values obtained from Kinect Azure.

Seven body segments can be extracted from the mannequin: (i) head and trunk, (ii) left forearm, (iii) right forearm, (iv) left arm, (v) right arm, (vi) left hand, (vii) and right hand. According to [43] and by considering the mannequin adopted to evaluate the human body kinematics, six absolute angles of interest should be considered:

$\alpha_{L}$: angle between the vertical axis and the left forearm.

$\alpha_{R}$: angle between the vertical axis and the right forearm.

$\beta_{L}$: angle between the left forearm and the left arm.

$\beta_{R}$: angle between the right forearm and the right arm.

$\gamma_{L}$: angle between the left arm and the left hand.

$\gamma_{R}$: angle between the right arm and the right hand.

The cameras were placed about 2 m from the mannequin, according to their FoV. The mannequin was moved in seven different poses, as shown in Figure 9, simulating a variety of human postures. For each configuration, a single point cloud was acquired. Points belonging to the mannequin were extracted using PolyWorks from each point cloud. Then, for each body segment, the Principal Component Analysis (PCA) was performed to compute the direction of their principal components considering the body axial symmetry [53,56]. Finally, the angles of interest were obtained by performing the scalar product between the principal components accordingly.

In addition to the reference values obtained from the Konica Minolta, the results were compared with the values obtained by Kinect v2. Figure 10 shows the resulting values of the angles of interest for each mannequin configuration. From these bar plots, it is evident that Basler ToF 640 (red bar) and Kinect v2 (orange bar) are the ones that achieve the worst performance. In contrast, Basler Blaze 101 (green bar) and Azure Kinect (blue bar) show a very similar trend with Konica Minolta (purple bar, golden standard). Excluding the performance of Kinect v2, the easiest poses to measure are poses 4 and 5, while for poses 1, 2, and 3, the three cameras show values that are sometimes very different from each other. This depends on the pose itself: poses 1, 2, and 3 span along the *z*-axis more than the other, resulting in higher occlusions.

To better analyze the results, the differences between the resulting values and the corresponding angle measured by the Konica Minolta are shown in bar plots in Figure 11. For most configurations, the Kinect v2 measurements are noticeably different from the reference ones acquired with Konica Minolta, sometimes reaching values of more than −30 deg (measured angle higher than reference) and more than 50 deg (measured angle lower than reference). The $\beta_{L}$ angle measured by Basler ToF 640 is the most different from the reference for all configurations, achieving errors ranging from more than −20 deg up to more than 30 deg. In comparison, Basler Blaze 101 performs better, achieving errors of −10 deg up to 25 deg. Azure Kinect measurements are typically the same as those computed from Konica Minolta, with errors ranging from −10 deg up to 12 deg, mostly occurring in correspondence of angles $\gamma_{L}$ and $\gamma_{R}$. These two angles consider the arm and the hand segments, which are difficult to measure accurately since the point clouds are less dense. These tests showed that Kinect Azure’s performance in reconstructing human bodies is notably improved compared to Kinect v2, resulting in lower errors. It is worth noting that this analysis has been performed without the aid of the Kinect Azure SDK, which computes the human skeleton using a skeletonization algorithm. Instead, the body segments have been extracted from the point clouds by using PolyWorks and approximated as the principal component vector resulting from the PCA analysis. As a result, each body segment was considered as a vector facilitating the estimation of angles between them. 

## 6. Conclusions

This paper details an experimental procedure to assess the metrological characteristics of ToF cameras with respect to the typical error sources of this technology. This contribution is especially important to robustly compare results among devices. The procedure was conducted on a consumer-end device (Kinect Azure) and two industrial ones (Basler Blaze 101 and Basler ToF 640). Although comparing cameras belonging to different worlds may seem counter-intuitive, this choice is motivated by the necessity of new solutions for modern industry, which is moving towards innovative environments where both humans and machine collaborate.

Error sources such as temperature influence on depth measurement, depth distortion, depth amplitude errors, temporal errors, and overall depth measurement uncertainty were evaluated for the three devices in different experiments. A summary of the developed evaluation procedure can be found in Table 2 for quick reference. The results of each camera are summarized in Table 3 in comparison with the corresponding datasheet and relevant references.

The presence of flying pixel problems and multipath errors was observed in the point cloud acquisitions; however, they were not quantified in this study. From these results, we may conclude that the best and worst performing cameras among the three are Kinect Azure and Basler ToF 640, respectively, while Basler Blaze 101 achieves comparable results with respect to Kinect Azure.

Since the metrological characterization of error sources may not be enough to determine the right device for a target application, we proposed an example in which the cameras are used for 3D reconstruction. The first experiment involved a cylindrical object, and the aim was to correctly estimate its diameter from the point cloud at different heights and distances from the cameras. However, from this test, the point clouds of the two industrial cameras resulted in being too noisy to be acceptable for evaluation. This was probably because they are best suited for objects with regular shapes placed at shorter distances from the light emitter. It is worth mentioning that from the error source characterization alone, it was not possible to predict such inconsistent behavior, thus highlighting the need for a standard experimental procedure for the assessment of ToF cameras’ 3D reconstruction capabilities.

The second experiment was aimed at estimating the capability of the cameras to reconstruct human bodies. This is especially useful for healthcare applications and for human activity monitoring and safety in industrial workspaces. The target object was a mannequin representing the human upper body with movable arms to simulate a variety of human poses. The reconstruction results of each camera were compared with Kinect v2 and with an industrial digitizer Konica Minolta (gold standard). In conclusion, the performance of Kinect Azure and Basler Blaze 101 are usually comparable with the reference except for some tricky poses where occlusions interfere with the measurement. The difference between the two is probably due to the typical application for which the cameras are developed. Industrial cameras perform best with smaller objects and have higher point cloud density to better reconstruct surface defects in controlled environments, while consumer-end cameras are typically used in unstructured environments with a variety of ambient conditions to reconstruct bigger objects and bodies.

As a further development, we aim to expand the characterization procedure by adding tests on different surfaces in terms of both materials’ opacity and color. Moreover, the 3D reconstruction experimental procedure proved to be necessary as well to define which camera is best suited for the task; hence, we aim to rigorously standardize it as well in the future by also taking into consideration objects of different shapes and materials, a full-body mannequin, and human subjects. In this way, researchers and practitioners may conduct a thorough metrological investigation of their sensor of choice.

## Figures and Tables

**Figure 1 sensors-23-00538-f001:**
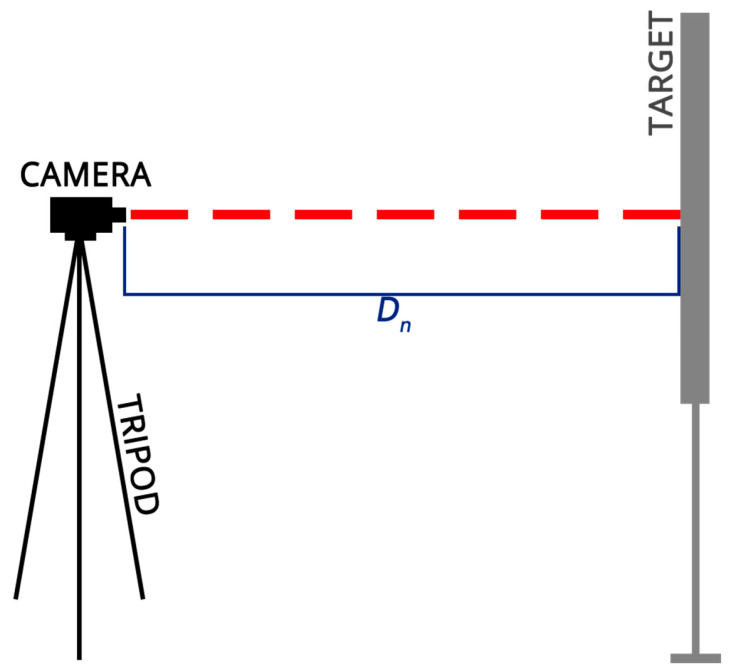
Scheme of the measuring set-up.

**Figure 2 sensors-23-00538-f002:**
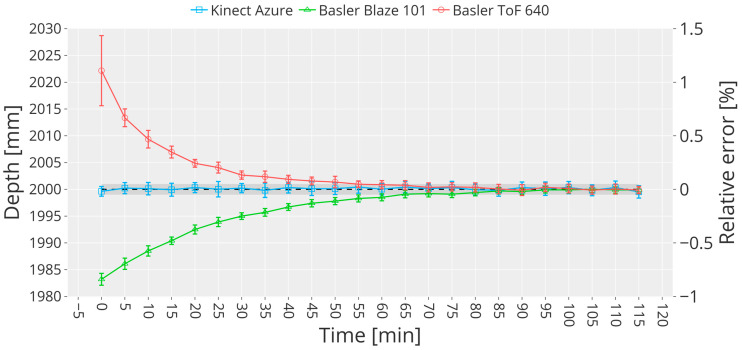
Temperature-related error. The plot shows the depth measurements obtained from the three ToF cameras considering a time interval of 5 min. Bias correction applied.

**Figure 3 sensors-23-00538-f003:**
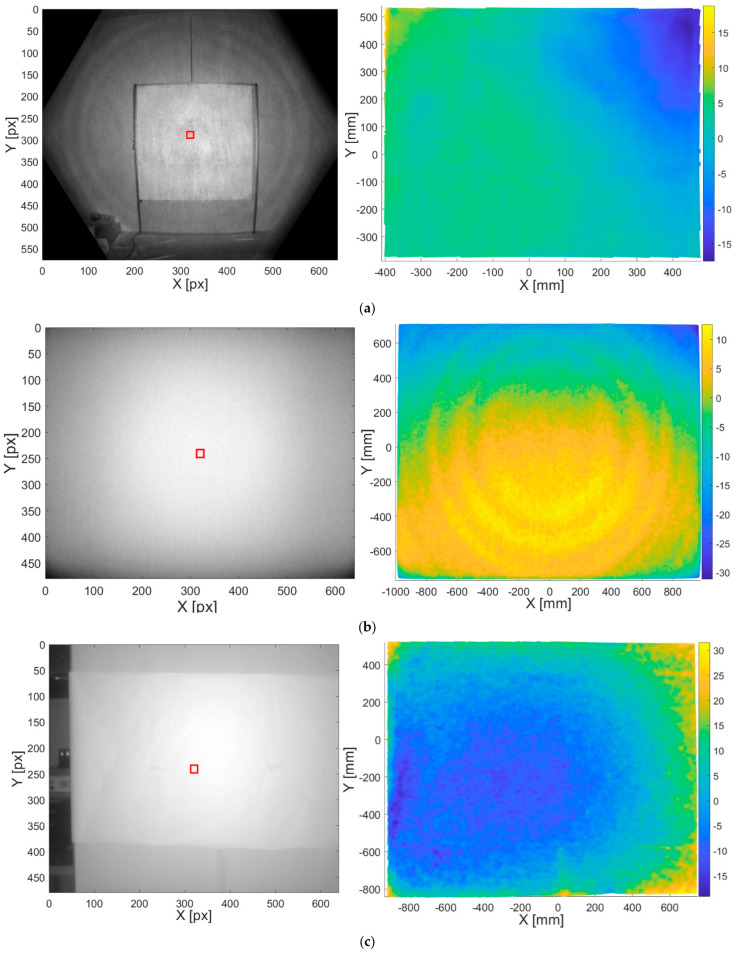
IR and Depth amplitude images of the target at $D_{n}=1.7\;$m acquired with (**a**) Kinect Azure, (**b**) Basler Blaze 101, (**c**) Basler ToF 640. The central region of interest is shown in red.

**Figure 4 sensors-23-00538-f004:**
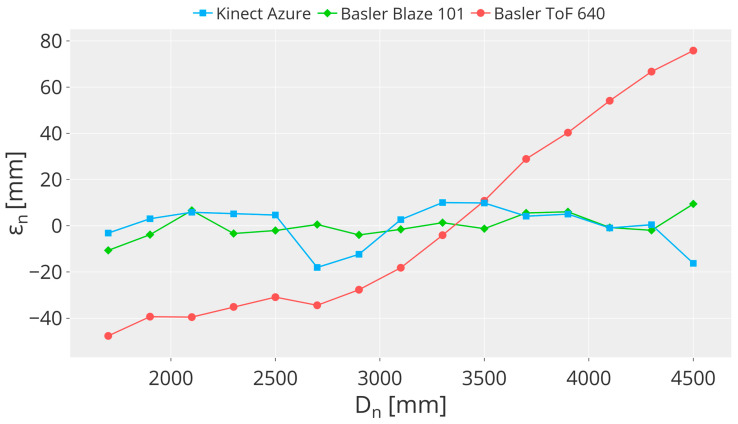
Plot showing the depth distortion error $\varepsilon_{n}$computed for each camera for each $D_{n}$.

**Figure 5 sensors-23-00538-f005:**
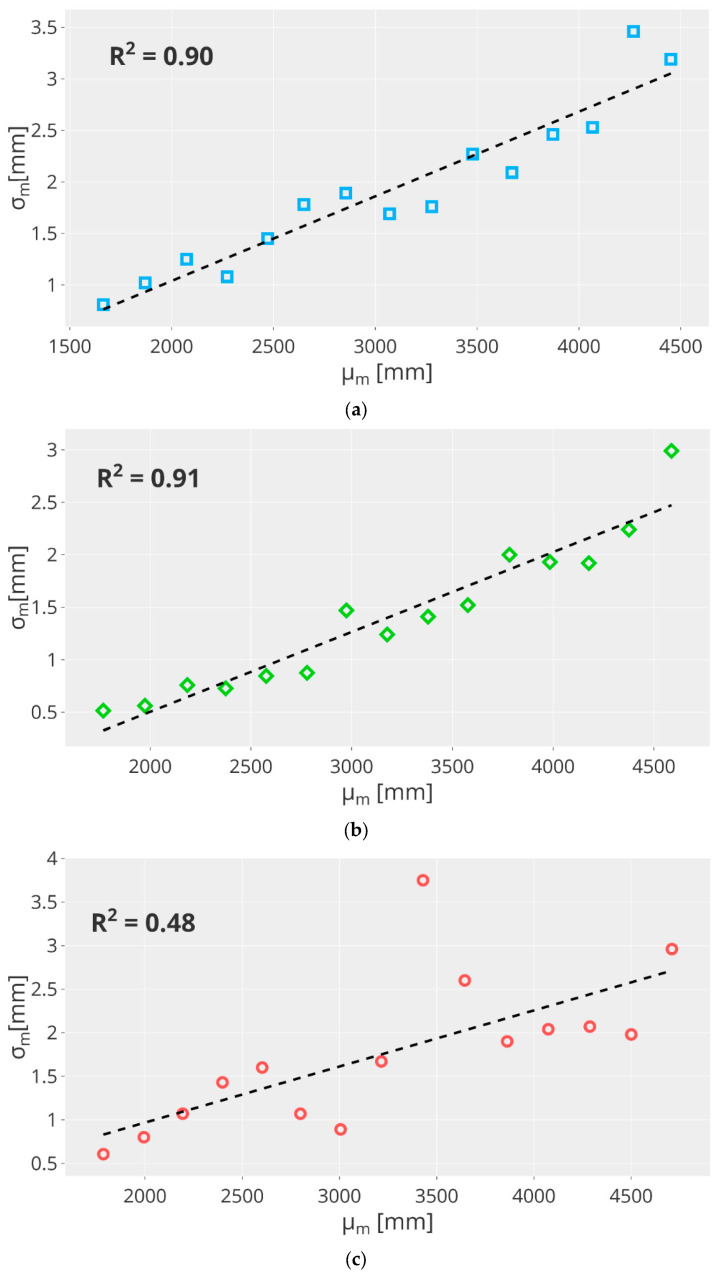
Plot showing the temporal error as a function of depth for (**a**) Kinect Azure, (**b**) Basler Blaze 101, and (**c**) Basler ToF 640.

**Figure 6 sensors-23-00538-f006:**
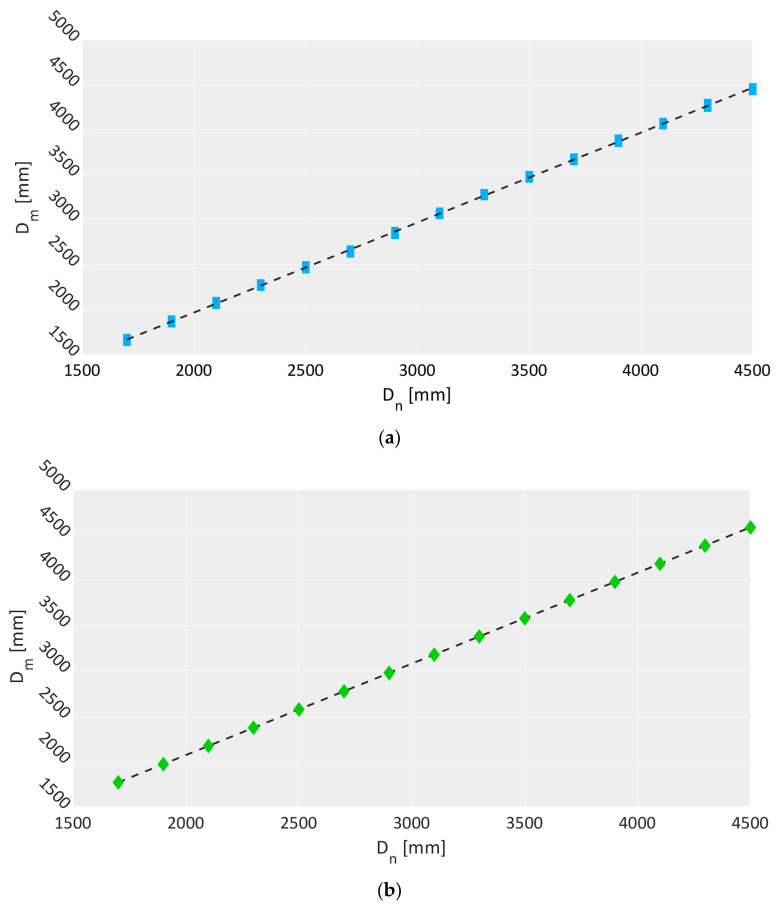
Measured depths $D_{m}\;$as a function of nominal depths $D_{n}$ for (**a**) Kinect Azure, (**b**) Basler Blaze 101, (**c**) Basler ToF 640.

**Figure 7 sensors-23-00538-f007:**
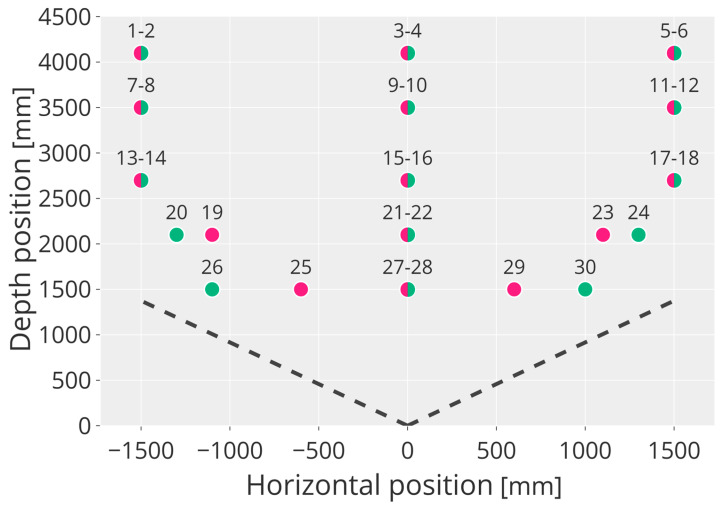
Cylinder positions adopted in the experiment. Red dots refer to odd station numbers, green dots to even station numbers.

**Figure 8 sensors-23-00538-f008:**
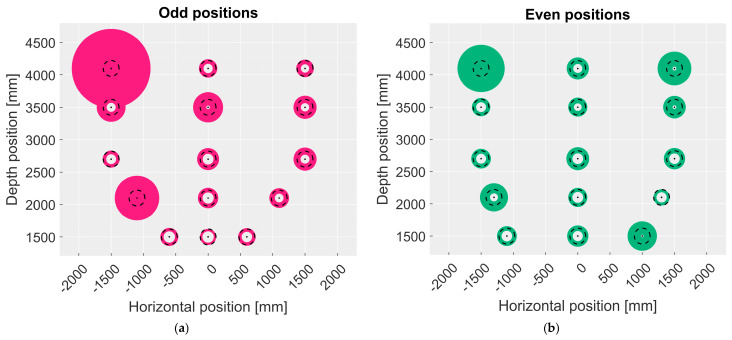
Image showing the upper bound of the cylinder’s diameter as the dimension of the colored circle, and the lower bound as the dimension of the white circle that results for each odd station. The dashed black circle represents the reference diameter. (**a**) Odd positions, (**b**) even positions.

**Figure 9 sensors-23-00538-f009:**
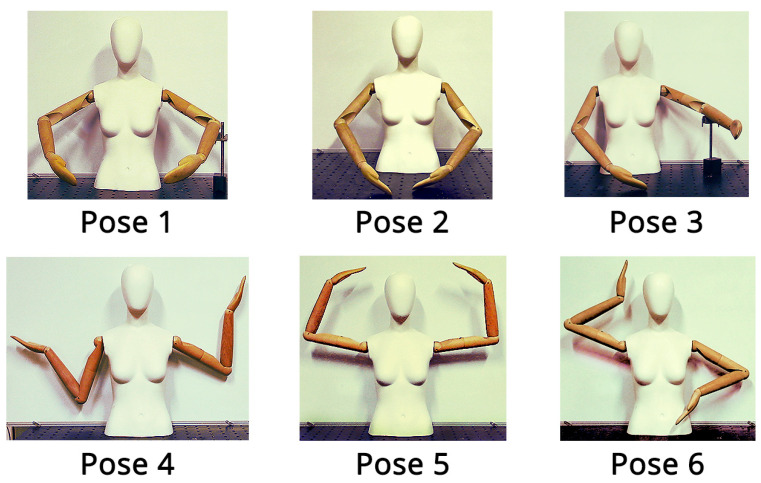
Dummy poses considered for the analysis as in [43].

**Figure 10 sensors-23-00538-f010:**
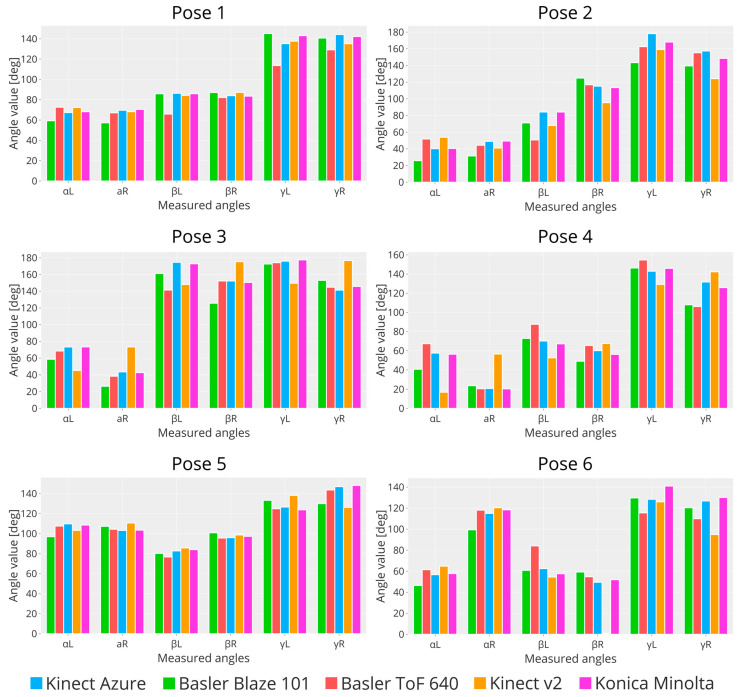
Graphs showing the measured angles of interest for the six poses.

**Figure 11 sensors-23-00538-f011:**
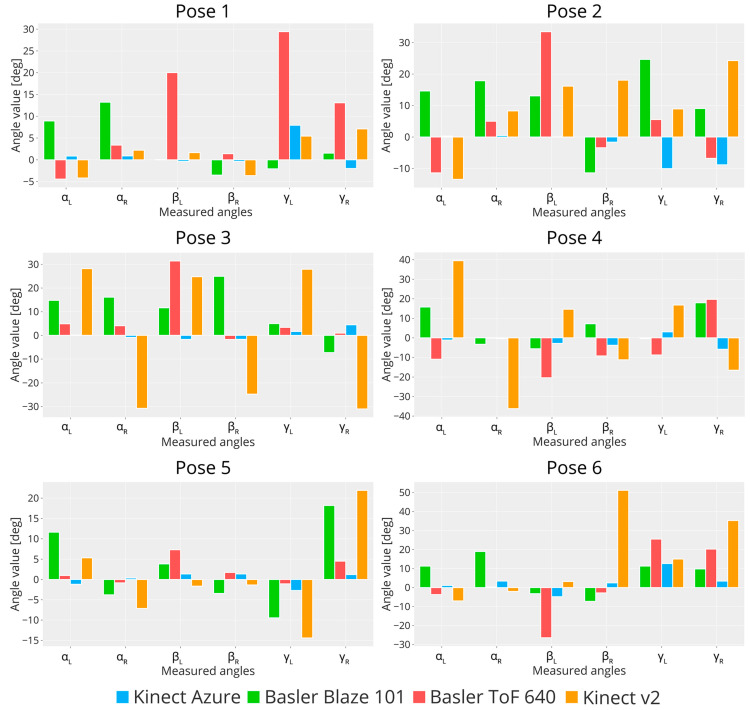
Bar plots representing the measuring error achieved by the three sensors and the old Kinect v2 for the six poses considered. The error is computed by subtracting the angle value obtained from Konica Minolta from the camera’s measure.

**Table 1 sensors-23-00538-t001:** Summary of cameras’ depth mode technical characteristics.

	Kinect Azure NFOV Unb.	Basler ToF 640	Basler Blaze 101
**Resolution**	640 × 576 px	640 × 480 px	640 × 480 px
**Frame rate**	30 fps	20 fps	30 fps
**FoV**	75 × 65 deg	57 × 43 deg	67 × 51 deg
**Working range**	0.5–3.86 m	0.5–5.8 m	0.5–5.5 m
**Dimension**	103 × 39 × 126 mm	141.9 × 76.4 × 61.5 mm	100 × 81 × 64 mm
**Power**	5.9 W	15 W	22 W
**Weight**	0.440 kg	0.400 kg	0.690 kg

**Table 2 sensors-23-00538-t002:** Summary of the proposed ToF camera evaluation protocol.

	Warm-Up Time	Depth Amplitude	Depth Distortion	Temporal Error	Overall Uncertainty
**Env. conditions**	Ensure optimal temperature (i.e., 24 °C) Ensure constant illumination without natural light interference				
**Reference target**	Opaque target with verified planarity especially in the central region				
**Hardware set-up**	Camera mounted on support at fixed height Ensure camera perpendicularity with respect to the target				
	Fixed distance $D_{n}$ Turn off camera before experiment for at least 4 h	Define a set of distances $D_{n}$ in the optimal working range according to the camera datasheet			
**Data acquisition**	1 depth frame or point cloud every 10 s at 30 fps	30 depth frames or point cloud at each $D_{n}$ at 30 fps			
**Data analysis**	Group frames belonging to 5 min time windows (30 frames total) Extract 15 × 15 ROI around the central pixel Compute mean depth $\mu_{t}$ and standard deviation $\sigma_{t}$ (Equation (1))	Extract target ∀ frame ∈ to each $D_{n}$ Compute error $\varepsilon_{n}$ (Equations (2)–(5))	Extract only the depth value of the central pixel ∀ frame ∈ to each $D_{n}$ Compute error $\varepsilon_{n}$ (Equations (2)–(5))	Extract only the depth value of the central pixel ∀ frame ∈ to each $D_{n}$ Compute deviation $\sigma_{m}$ (Equation (6)) Compute linear regression and check $R^{2}$	Extract 20 × 20 ROI around the central pixel Use all data points inside ROI Compute linear regression and check $R^{2}$
**Data correction**	Apply bias correction and obtain $d_{t}^{*}$ (Equation (1))	Ensure that $\varepsilon_{n}$ is the relative error not the absolute depth		//	Remove outliers before applying linear regression
**How to visualize**	*X*-axis: time [s] *Y*-axis: $d_{t}^{*}$ with corresponding $\sigma_{t}$ [mm] Optional: secondary *y*-axis showing relative error [%]	IR image and Depth error *X*-axis: x coordinate [px] and [mm], respectively *Y*-axis: y coordinate [px] and [mm], respectively Show color bar	*X*-axis: distance $D_{n}$ [mm] *Y*-axis: $\varepsilon_{n}$ [mm]	*X*-axis: $\mu_{m}$ [mm] *Y*-axis: $\sigma_{m}$ [mm] Show linear regression line	*X*-axis: $D_{n}$ [mm] *Y*-axis: $D_{m}$ [mm] Show linear regression line

**Table 3 sensors-23-00538-t003:** Summary of the error sources influence resulting from our experiments in comparison with relevant references and cameras’ datasheets. Asterisks refer to data obtained by our procedure. DS refers to data found in the device’s datasheet.

	Warm-Up Time	Depth Amplitude	Depth Distortion	Temporal Error	Overall Uncertainty
**Kinect Azure ***	Not needed	2 to 5 mm	−18 to 10 mm	0.8 to 3.2 mm	13 mm
**Kinect Azure DS**	Not provided	Not provided	$\text{<}11\;\mathrm{mm}+0.1\%\;D_{n}$	≤17 mm	Not provided
**Kinect Azure** [1]	50 min	Not provided	−7 to 0 mm	0.5 to 2 mm	Not provided
**Kinect Azure** [2]	Not provided	−2 to 0 mm	1.1 to 12.7 mm	0.6 to 3.7 mm	Not provided
**Basler Blaze 101 ***	50 min	3 to 7 mm	−11 to 9 mm	0.5 to 3 mm	6 mm
**Basler Blaze 101 DS**	20 min	−5 to 5 mm	Not provided	<2 mm	Not provided
**Basler ToF 640 ***	50 min	−10 to −6 mm	−48 to 76 mm	0.6 to 3.8 mm	13 mm
**Basler ToF 640 DS**	20 min	−10 to 10 mm	Not provided	≤8 mm	Not provided

## Data Availability

The data presented in this study are available on request from the corresponding author. The data are not publicly available due to privacy.

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
