# Peer review of "Experimental Procedure for the Metrological Characterization of Time-of-Flight Cameras for Human Body 3D Measurements"

_sensors, 2023, doi:10.3390/s23010538_

Round 1

Reviewer 1 Report

Dear Authors,
Thank you for your paper on an experimental procedure for the metrological characterization of time-of-flight cameras for human body 3D measurements.

The procedure, the methods, and the devices are presented from the right point of view, in my opinion.

The experimentation permits an evaluation of the proposed method.

I have not clear the scientific news of your method. These results seem to be a technical review of a specialized newspaper.

Please put in evidence the scientific news you introduce with this work.

1.Regarding the methodology, authors should show what the innovative aspect of the problem they are facing is.

2. The conclusion paragraph shows mainly the technical characteristics of the cameras. It is a conclusion of a technical report, not of a scientific paper.

3.  The references are appropriate.

4. The authors should put in evidence the innovation presented of this paper. The paper presents a mix of a technical review (good for a magazine) and a measurement method.  Perhaps the most interesting argument is the methodology, but the authors do not develop this argument. After a brief introduction to this methodology, the starts with the technical review of the cameras.

The authors should provide a deep description of the method they introduce, providing the motivation for the new method and mathematic properties (like statistical advantages).

After e formal description of the method, they could show an application of the method with these tests. At the moment, the paper is too much confused, and the objective of the research is not clear.

Perhaps they could write two papers: the first one that describes the method and the second one with an application of the method. 

Regards

Author Response

Thank you for your review. Please see attachment.

Reviewer 2 Report

The paper suggests a new technique to evaluate the characteristics of ToF cameras with respect to typical errors.

The authors use the word "Meteorology" several times. Meteorology is a science that deals with the study of atmospheric phenomena, weather processes and predictions. Meteorology belongs to earth sciences. It is not the correct word for this paper.

It would be nice to detail the equations that the authors used in order to choose the number they have used in the paper.

The use of cameras in order to automatically classify objects and how to deal with possible errors was suggested in several cases like detecting damaged vehicle tires in Y. Wiseman, "Take a picture of your tire!," Proceedings of 2010 IEEE International Conference on Vehicular Electronics and Safety, 2010, pp. 151-156 available online at: https://u.cs.biu.ac.il/~wisemay/icves2010.pdf  and also identifying chicken eggs suitable for incubation in Fernández-S., Á., Salazar-L., F., Jurado, M., Castellanos, E.X., Moreno-P., R., Buele, J. (2019). Electronic System for the Detection of Chicken Eggs Suitable for Incubation Through Image Processing. In: Rocha, Á., Adeli, H., Reis, L., Costanzo, S. (eds) New Knowledge in Information Systems and Technologies. WorldCIST'19, 2019. Advances in Intelligent Systems and Computing, vol 931, Springer. I would encourage the authors to cite these two papers and explain how their suggestion can enhance the use of cameras for these purposes.

Author Response

(The authors gave the same response as above.)

Round 2

Reviewer 1 Report

Dar Author,

The paper is accepted.

Regards

Author Response

Thank you for your reveiw.

Reviewer 2 Report

The authors made only some superficial changes and this is actually almost the same version. I would encourage the authors to revise the manuscript thoroughly.
